# Signaling Induced by Chronic Viral Hepatitis: Dependence and Consequences

**DOI:** 10.3390/ijms23052787

**Published:** 2022-03-03

**Authors:** Zakaria Boulahtouf, Alessia Virzì, Thomas F. Baumert, Eloi R. Verrier, Joachim Lupberger

**Affiliations:** 1Institut de Recherche sur les Maladies Virales et Hepatiques UMR_S1110, Université de Strasbourg, Inserm, F-67000 Strasbourg, France; zakaria.boulahtouf@etu.unistra.fr (Z.B.); virzi@unistra.fr (A.V.); thomas.baumert@unistra.fr (T.F.B.); e.verrier@unistra.fr (E.R.V.); 2Service d’Hépato-Gastroentérologie, Hôpitaux Universitaires de Strasbourg, F-67000 Strasbourg, France; 3Institut Universitaire de France (IUF), F-75005 Paris, France

**Keywords:** HBV, HCV, HDV, liver, inflammation, oxidative stress, metabolic disease, fibrosis, cancer

## Abstract

Chronic viral hepatitis is a main cause of liver disease and hepatocellular carcinoma. There are striking similarities in the pathological impact of hepatitis B, C, and D, although these diseases are caused by very different viruses. Paired with the conventional study of protein–host interactions, the rapid technological development of -omics and bioinformatics has allowed highlighting the important role of signaling networks in viral pathogenesis. In this review, we provide an integrated look on the three major viruses associated with chronic viral hepatitis in patients, summarizing similarities and differences in virus-induced cellular signaling relevant to the viral life cycles and liver disease progression.

## 1. Introduction

Viral hepatitis predominantly affects and damages the liver by commonly causing the progression from chronic inflammation to fibrosis, cirrhosis, and ultimately cancer. It is estimated that approximatively 350 million people worldwide are chronically infected with hepatitis viruses [1]. During a chronic infection of the liver, hepatic viruses persistently tweak and attenuate the host antiviral defenses and modulate cellular pathways that impact liver homeostasis and disease progression. Viral hepatitis is a major risk factor for liver fibrosis, cirrhosis, and hepatocellular carcinoma (HCC), which is the second leading and fastest rising cause of cancer death worldwide [2,3]. Although caused by very different viruses, virus-induced liver disease displays similar features, suggesting common molecular drivers. Moreover, chronic viral hepatitis may serve also as a model to understand the mechanism of non-viral etiologies like non-alcoholic fatty liver disease (NAFLD) and non-alcoholic steato-hepatitis (NASH).

Although replication strategies of hepatotropic viruses are very diverse, we can highlight common molecular mechanisms that occur during chronic viral hepatitis: (1) induction of intrahepatic oxidative stress damage by viral proteins, (2) dysregulation of cellular metabolic pathways, (3) persistence of liver inflammation, (4) activation of pro-fibrotic, pro-oncogenic processes that can lead to the accumulation of genetic alterations and genomic instability. Therefore, the common denominator of these events comprises a virus-induced dysregulation of signaling events that holds the potential for the identification of novel host-targeting and chemo-preventive strategies targeting the viral life cycle and/or liver disease progression. In this review, we summarize similarities and differences in virus-induced cellular signaling associated with the three major viruses that cause chronic viral hepatitis in patients.

## 2. Hepatitis B, C, D Viruses

The three hepatotropic viruses causing chronic liver infection are Hepatitis B virus (HBV), Hepatitis C virus (HCV), and Hepatitis Delta virus (HDV). HBV is a DNA virus of the *Hepadnaviridae* family whose partially double-stranded genome is translocated into the host nucleus. Here, a covalently closed circular DNA (cccDNA) is formed and serves as a template for transcription. cccDNA is highly persistent and epigenetically regulated as a host chromosome, a critical feature that makes it difficult to achieve a complete cure for HBV infection [4,5]. HBV is the only hepatotropic virus that causes integration of the viral DNA into the host genome. It thus contributes directly to an elevated liver cancer risk even in non-cirrhotic patients by cis-mediated insertional mutagenesis, chromosomal instability, and expression of aberrant viral proteins [6]. HDV is a single-stranded RNA virus and the only virus of the genus *Deltavirus*. It is a satellite virus of HBV, which requires the HBV surface antigen (HBsAg) for its lifecycle [7,8]. Importantly, HBV/HDV coinfection causes the most severe form of chronic viral hepatitis, with accelerated liver disease progression to cirrhosis and HCC and increased liver-related and overall mortality [9]. However, little is known about the HBV/HDV–host interactions driving these complications. HCV, a member of the *Flaviviridae* family, is a positive-sensed single-stranded RNA virus that depends on and interacts with hepatocyte lipid metabolism during its lifecycle [10]. HCV triggers phenotypic changes closely resembling metabolic liver disease, including hepatic steatosis and insulin resistance [11], and profoundly influences the proteogenomic landscape of the host cell [12].

Challenges in the treatment of these viruses differ very much. While an efficient HBV vaccine is available protecting from HBV and HDV, the development of an HCV vaccine is hampered by its lipoviral composition and its highly variable quasispecies that contribute to its shielding and escape from neutralizing antibodies. However, over the last decade, novel and highly efficient antivirals have been developed to cure HCV infection [13]. In contrast, chronic HBV infection can only be controlled by long-term antiviral strategies due to the persistence of cccDNA pools in patients’ liver [14,15]. For HDV, interferon-based therapies had only limited success; however, novel antivirals such as entry inhibitors have shown encouraging results in clinical practice to control HDV infection and to improve liver function [16]. Nevertheless, even if a viral infection is controlled or cured, the risk of developing HCC may not be fully reversed, depending on the duration of chronic infection, liver disease stage, and type of virus [17,18]. Moreover, evidence points towards an epigenetic imprinting by hepatic viruses (HCV, HBV) and underlying liver fibrosis in the host genome which maintains a persistent transcriptomic environment in cured livers that acts in a pro-oncogenic manner [17,19,20].

## 3. Virus-Induced Oxidative Stress Signaling

In healthy cells, reactive oxygen species (ROS) are predominantly produced through mitochondria oxidative phosphorylation, protein folding in the endoplasmic reticulum (ER), and the catabolism of lipids and amino acids [21,22,23]. ROS are considered to be harmful for the cell, exerting damage-promoting, detrimental effects. However, ROS are also an essential signaling trigger regulating apoptosis and immune response against pathogens [24]. ROS are neutralized by the enzymatic and non-enzymatic cellular antioxidant system. The enzymatic antioxidant system includes various types of ROS-scavenging phase II enzymes such as glutathione peroxidase (GPx), superoxide dismutase (SOD), and catalase (CAT) catalyzing free radicals’ neutralization. In contrast, the non-enzymatic antioxidants system is composed of low-molecular-weight compounds such as glutathione and vitamin C, scavenging ROS with a slow kinetics. Both systems are regulated by the expression of genes comprising antioxidant response elements (ARE), which are controlled by the transcription factor Nrf2 [25,26]. A persistent imbalance of ROS is an important driver of chronic liver disease, and the associated redox imbalance has been suggested to be highly relevant to NAFLD pathogenesis [27]. Moreover, oxidative stress has been associated with oncogenic transformation in patients with chronic viral hepatitis [28,29].

The mechanism of virus-induced oxidative stress by HBV, HCV, HDV can be summarized in four main categories: (1) alteration of mitochondrial function mediated by Ca^2+^ uptake; (2) triggering of ER stress and unfolded protein response (UPR); (3) virus-induced expression of ROS-producing enzymes; (4) dysregulation of antioxidative pathways (Figure 1). In the case of HCV, viral core proteins [30], E1/E2 [31], and NS4B [32,33] induce oxidative stress via calcium efflux through the induction of ER stress and UPR, which is a component of the ER adaptative system. In addition, the HCV core at the mitochondrial outer membrane [34] interacts with heat shock protein Hsp60 [35], triggering the release of Ca^2+^ from the ER and its accumulation in the mitochondria. Moreover, the HCV core stimulates the expression oxidoreductin 1α (ERO1α) in the ER. This promotes the formation of mitochondria-associated membranes and induces Ca^2+^ translocation from the ER to the mitochondria. Mitochondrial Ca^2+^ accumulation alters the respiratory chain and promotes ROS production [36,37]. For HBV, HBx protein expression reduces the activity of several respiratory chain complexes, causing the loss of mitochondrial membrane potential and therefore enhancing the production of ROS [38]. Moreover, HBx dysregulates mitochondrial functioning by interacting with two partners: voltage-dependent anion channel 3 (VDAC3), involved in calcium transport across the mitochondrial outer membrane [39], and cytochrome c oxidase subunit III (COX3) [40,41]. Other HBV proteins may also be involved in the induction of oxidative stress. HBsAg is generally secreted during the HBV lifecycle; however, secretion-deficient mutants can appear during infection and accumulate in the ER. This also occurs with the HBV core antigen (HBcAg). Both proteins induce ER stress and UPR signaling, leading to calcium release from the ER and subsequent ROS production [42,43]. Viral components also enhance oxidative stress by inducing the expression of ROS-producing enzymes. The HCV core proteins and NS5A enhance the expression of cytochrome P450 2E1 (CYP2E1) and NADPH oxidase 1 and 4 (NOX1 and 4), leading to elevated levels of ROS, including superoxide and hydrogen peroxide [37,44]. Similarly, the large HDV surface antigen (L-HDAg) induces oxidative stress by promoting NOX4 expression [45].

A virus-induced, but not always consistent, dysregulation of the antioxidant system is observed during HCV and HBV infection. For HCV, the expression of nonstructural proteins downregulates SOD1 and SOD2 and induce catalase, whereas HCV core alone enhances the expression of SOD2 [46]. The expression of full-length HCV or the nonstructural proteins and core leads to impaired Nrf2/ARE activity [47], whereas a full HCV infection activates the Nrf2/ARE axis [48,49,50]. Overexpression studies in Huh7 cells point towards Nrf2-activating phosphorylation during HCV infection by protein kinase C in response to ROS or by casein kinase 2 and phosphoinositide 3-kinase (PI3K) in a ROS-independent manner [48]. Moreover, in an HCV infection model, Nrf2/ARE activation was promoted by the inhibitory phosphorylation of glycogen synthase kinase 3β (GSK3β) [49]. While in non-transformed hepatocytes several Nrf2-dependent genes are induced by HCV [51], downregulation of a wide spectrum of antioxidant defense proteins can be observed in hepatoma cell line-based models [52,53]. Several theories have been proposed to explain this discrepancy, one of which is the bi-phasic nature of oxidative stress, which at low and moderate ROS levels activates antioxidants, whereas at high ROS level induces damage and inhibits the expression of antioxidant genes. The used study model and the readout have thus a significant impact on the results. HCV-induced gene expression is often not translated into a protein response to blunt the antiviral response of the host cells [12]. For HBV, the Nrf2/ARE pathway is activated by the virus in both infected cells and liver tissues of chronic HBV carriers in a genotype-dependent manner via HBx and the large surface antigen (LHBs) [54]. HBx sequesters the Nrf2 partner protein, Keap1, forming a HBx–p62–Keap1 triple complex in a ROS-independent manner [55]. However, the activation of the antioxidant system by HBV is challenged by several studies in HBV-infected cells and in HBV patients [56,57,58]. Indeed, some Nrf2-dependent genes such as *GSTM3* [59] and *GSTP1* [60] are epigenetically silenced by HBx expression or HBV infection. HBx also alters type II enzyme expression by interfering with the expression of other regulatory elements/factors of the Nrf2/ARE signaling pathway [61]. Furthermore, HBV suppresses the expression of proteins indirectly implicated in the antioxidant system such as selenoprotein P (SeP) and selenium-binding protein 2 (Selenbp2) [62,63]. Beside the observed effects of HBV on ROS, also HDV promotes oxidative stress in the ER through the interaction between L-HDAg and NOX4. The activation of the NOX4 pathway induces the release of ROS from the ER, activating the signal transducer and activator of transcription-3 (STAT3) and nuclear factor kappa B (NF-κB) signaling [45]. Moreover, the small hepatitis delta antigen (S-HDAg) can directly bind to glutathione S-transferase P1 mRNA causing the downregulation of its expression, therefore increasing ROS and promoting apoptosis [64].

Oxidative stress, ER stress, and UPR trigger a cascade of signaling events that may protect but also damage the liver, depending on the duration of the insult. During HCV infection, elevated ROS levels induce the production of pro-inflammatory cytokines such as tumor necrosis factor alpha (TNF-α) and IL-8 [65]. The underlying NF-κB pathway regulating their production is very sensitive to oxidative stimuli. HCV core proteins, NS4B, and NS5A activate NF-κB as well as STAT3 expression through the elevation of ROS and the disruption of calcium homeostasis [66,67]. Interestingly, the induction of oxidative stress during HCV infection is positively associated with the progression of liver fibrosis, which is characterized by the overproduction of extracellular matrix. A key mechanism of fibrosis is the activation of hepatic stellate cells (HSCs) by pro-inflammatory cytokines, leading to collagen deposition [68]. HCV core proteins, NS3A/4A, NS4B, and NS5A activate transforming growth factor beta 1 (TGF-β1) secretion through ROS and calcium-dependent mechanisms [69]. Viruses also benefit from manipulating ROS levels. Studies revealed that HCV activates the Nrf2/ARE axis, promoting ROS scavenging and preventing ROS accumulation to levels with antiviral and/or lethal effects in the host cell [48,70]. Moreover, under conditions of oxidative stress, viruses promote cell survival and proliferation via associated signaling. HCV activates β-catenin that induces c-Myc and cyclin D1 expression, thus promoting cell cycle progression [71]. Moreover, ROS disrupt p53 binding to Mdm2 via the upregulation of DHCR24 expression, thus attenuating apoptosis [72]. HCV further prevents apoptosis by activating the peroxisome proliferator-activated receptor alpha (PPARα) and suppressing the voltage-gated K^+^ channel Kv2.1 through NS5A and ROS [73,74]. ROS also suppress the expression of p14, which is implicated in the induction of the pro-apoptotic p53/Mdm2 pathway [75]. ROS upregulates p21, a cyclin-dependent kinase inhibitor activating Nrf2 [76]. While in low stress conditions p21 induces cell cycle arrest, in the presence of high oxidative stress levels, it induces apoptosis [77,78]. HCV core proteins and NS5A inhibit p21 and therefore render Nrf2 less sensitive to ROS. This hampers the induction of apoptosis and stimulates the proliferation of damaged hepatocytes [79,80].

## 4. Virus-Induced Pro-Inflammatory Signaling

A common consequence of chronic viral hepatis is the induction of liver inflammation (hepatitis). Upon viral sensing, infected hepatocytes trigger the activation of innate immune receptors and sensors that are referred to as the inflammasome and are large protein complexes. The inflammasome serves as a signaling hub triggering type I interferons and the processing and release of proinflammatory cytokines. It is activated by pattern recognition receptors (PRRs) triggered by pathogen-related molecular patterns (PAMPs) or damage-associated molecular patterns (DAMPs) (reviewed in more detail in [81]). The host inflammasome represents an important line of defense and is a decision maker with regard to fight (antiviral response) and containment (apoptosis). It is therefore not surprising that hepatic viruses evade the host innate immune response and twist pro-inflammatory signaling to their own benefit to persist and prevent apoptosis of the infected cells. The induction of the inflammasome by viral hepatitis was already evident in the previous section highlighting the common mechanism of virus-induced ROS and ER stress activating STAT3 and NF-κB signaling. Presumably, ROS-independent or -related mechanisms comprise the induction of NF-κB signaling by HDV. Indeed, L-HDAg renders NF-κB signaling more susceptible to TNF-α [82] and induces STAT3 [83]. NF-κB-independent induction of inflammation has been observed for HCV, which involves NS5A induction of cyclooxygenase-2 (COX-2) and, consequently, second messenger signaling and prostaglandin production [84]. The JAK/STAT signaling pathway is an important mediator of the host innate immune response as well as of cellular apoptosis and survival. Viral sensing triggers type I and II interferon responses via STAT1 and STAT2, promoting the expression of antiviral interferon response genes and apoptosis if the pathogen is not cleared [85]. Viruses causing chronic hepatitis have developed elaborated strategies to evade the innate response (reviewed in [86,87,88,89]). An important aspect in this evasion strategy is the pro-viral role of STAT3 signaling. In the liver, STAT3 signaling is a mediator of liver regeneration and balances the pro-apoptotic role of STAT1 by heterodimerization in response to IL-6 and proliferative signaling by HGF and EGF [90]. Attenuation of STAT3 signaling in functional studies impaired the replication of HCV and HBV, suggesting a pro-viral role of STAT3 signaling [90,91,92]. Proliferative signaling via the epidermal growth factor receptor (EGFR) is required for HBV, HCV, and HDV infection [93,94]. Interestingly, EGFR further promotes STAT3 activity by repressing a negative regulator of STAT3, i.e., suppressor of cytokine signaling 3 (SOCS3). Consequently, this tempers the pro-apoptotic, antiviral effect of type I interferon signaling by promoting STAT1/STAT3 heterodimerization over STAT1 homodimerization [95]. STAT3 signaling is further maintained by HCV-induced downregulation of another negative regulator of STAT3, protein tyrosine phosphatase delta (PTPRD) [96]. A potential similar effect may be triggered by HDV infection, which attenuates STAT1/STAT2 signaling via the suppression of Tyk2 [89], which is another negative regulator of STAT3 transcriptional activity [97]. Whether this may promote the pro-viral effect of STAT3 signaling in HBV/HDV-infected hepatocytes remains to be demonstrated.

## 5. Deregulation of Cellular Metabolism Pathways

The liver is an essential hub for metabolic processes and energy storage. Chronic viral hepatitis has an important impact on metabolic processes in the liver with distinct virus-specific manifestations. While chronic HCV infection strongly resembles clinical manifestations caused by NAFLD and NASH [98], the role of HBV in metabolic disease is controversial, and it has been suggested that HBV may potentially exhibit protective effects towards NAFLD development [99,100]. However, metabolic disease and obesity in HBV- or HCV-infected patients are considered important co-morbidities promoting liver disease progression and increasing cancer risk [101]. The accumulation of free fatty acids induces mitochondrial and ER oxidative stress. Moreover, the accumulation of ROS stimulates lipid peroxidation and inflammatory cascades such as those associated with TNF-α and IL-6, leading to the development of hepatic steatosis and insulin resistance [102].

HCV lifecycle is tightly linked to human lipid metabolism also because HCV requires lipid droplets to replicate and circulates as lipoviral particles to evade the host immune response [103]. HCV proteins directly interact with or regulate the expression of key effector molecules of the lipid metabolism, including apolipoproteins and diacylglycerol acyltransferase-1 [104,105,106]. HCV also activates IKK, which induces the expression of lipogenic genes and promotes lipid droplet formation [107]. Indeed, recent proteogenomic analysis revealed a massive suppression of pathways required for fatty acid metabolism, perturbing the capacity of infected hepatocytes to use fat as an energy source. This coincided with a shift towards a highly glutamine-/glucose-dependent metabolism, which promoted HCV replication [12,108] and resembled the high energy dependence of tumor cells. HCV also tweaks host signal transduction, promoting a favorable metabolic environment for its replication and persistence.

A central role in metabolic liver disease is a chronically dysregulated STAT3 and NF-κB signaling [109,110]. As reviewed above, both pathways are induced by chronic viral hepatitis and associated oxidative stress and are associated with liver disease progression and increased HCC risk [12,90,111,112,113,114]. However, while during HCV infection STAT3 signaling contributes to the accumulation of free fatty acids by suppressing peroxisomal beta-oxidation via inhibiting PPARα [12], HBV-induced STAT3 signaling does not produce the same phenotype. In contrast, HBV replication requires PPARα [115,116] and therefore prevents PPARα inhibition by inducing STAT3 activity. Indeed, HBV infection of primary human hepatocytes induces peroxisomal function, which may be a consequence of a direct rescue of PPARα activity by an HBV protein [12], as observed for PPARγ activation by the protein HBx [117]. HBx maintains fatty acid oxidation [118] and has been observed to bind PPARα in cell culture [119]. This matches observations that PPARα stimulation improves NAFLD in pre-clinical models [120,121]. However, evidence points also to a pro-steatotic impact of HBV infection, even though this is more rarely observed in patients compared to patients infected with HCV [122]. HBV infection does promote the biosynthesis of fatty acids in HBV transgenic (HBV-Tg) mice by the upregulation of fatty acid-binding protein 5 (FABBP5) and acyl-CoA-binding protein (ACBP) [123]. Moreover, HBV-Tg mice display the upregulation of lipid biosynthetic pathways such as those involving retinol-binding protein 1 (RBP1), sterol regulatory element-binding protein 2 (SREBP2), ATP citrate lyase, and fatty acid synthase (FAS) [124]. Additionally, other factors participating in fatty acid transport and biogenesis are dysregulated by HCV, including fatty acid-binding protein 1 (FABP1), responsible for the uptake and transport of long-chain fatty acids (LFA) [125,126], SREBP1, and PPARγ, which induces the expression of hepatic lipogenic and adipogenic genes, accompanied by the accumulation of lipid droplets [127,128,129,130]. HDV infection also impacts metabolic pathways, since HDV decreases the availability of triosephosphate isomerase and pyruvate carboxylase, leading to an abnormal retention of lipids. This effect may also be responsible for microvesicular steatosis during HDV infection [131].

A particular feature of HCV infection is its association with insulin resistance (IR) in patients, which is less frequently observed in HBV- or HBV/HDV-infected patients [132], although a recent genetic screen highlighted the importance of metabolic pathways in HDV lifecycle, including insulin resistance-related genes [133]. This may be due to the different dependency of the hepatic viruses on intracellular glucose levels. Insulin is a central regulator of glucose levels in the blood and of gluconeogenesis in the liver. It therefore also impacts the glucose levels within the hepatocyte. During IR, insulin fails to suppress gluconeogenesis in the hepatocytes [134]. While HCV is a highly glucose-dependent virus [12], HBV replication favors low glucose levels, and thus glucose-induced mTOR signaling hampers HBV replication [135]. HCV proteins directly promote IR by interacting with insulin pathways’ components, including insulin receptor substrate 1 (IRS-1) [136,137,138]. It also alters Akt-induced forkhead box O1 (FOXO1) phosphorylation and its nuclear exclusion, which is required for the transcription of the gluconeogenic gene phosphoenolpyruvate carboxykinase 1 (PCK1) in hepatocytes [139]. Moreover, HCV suppresses glucose transporter 2 (GLUT2) and IRS-2, contributing to higher endogenous glucose levels. Indirect mechanisms of IR involve HCV-induced oxidative stress, steatosis, and pro-inflammatory cytokines, e.g., TNF-α. These indirect effects induce the expression of gluconeogenic genes, such as glucose 6 phosphatase (G6P) and PCK2 [140,141]. Moreover, HCV mediates oxidative stress, leading to hypoxia. This activates H1Fα via c-Myc and Nrf2, controlling the expression of key enzymes in glycolysis [108]. One may speculate that HBV and HDV infection rather indirectly contribute to insulin resistance in patients via virus-induced inflammation and oxidative stress, which is further promoted by comorbidities such as overweight. Direct effects are observed, however, in HBx-Tg mice, which develop hyperglycemia and impaired glucose tolerance [142], and in the HBV-expressing cell line HepG2.2.15, with stimulated TCA cycle and glycolysis [143].

## 6. Virus-Induced Pro-Fibrotic/Pro-Oncogenic Signaling

Many of the above-mentioned dysregulated signaling pathways promote viral replication and persistence mostly by diverting the host antiviral response to prevent apoptosis and ensure the survival of the infected cell. Strikingly, many of these survival signals are also involved in regenerative processes during liver injury and orchestrate a delicate balance between pro-inflammatory and proliferative signals [144]. As mentioned earlier in this review, all three hepatis viruses chronically infecting the liver engage EGFR signaling to maintain their life cycle [93,94]. EGFR orchestrates the entry of HBV [93] and HCV [94]. EGFR signaling is active during ligand-induced receptor dimerization and internalization and is regulated by phosphatases and endosomal recycling/degradation [145]. HCV has developed strategies to maintain EGFR signaling to its own benefit. HCV infection induces EGFR signaling [146,147] and prolongs EGFR signaling by retaining EGFR in the early endosome via NS5A. This prevents EGFR degradation and leads to EGFR accumulation in infected cells [148,149]. HCV also alters the expression of other ErbB receptors in favor of EGFR [150]. In contrast to HCV, HBV internalization requires EGFR transport to the late endosome, which is critical for efficient HBV infection [151]. Consistently, the inhibition of EGFR degradation abrogated the internalization of HBV via its receptor sodium/taurocholate cotransporter (NTCP) and prevented viral infection [151]. Downstream of EGFR signaling, several viral proteins interact with the MAPK signaling pathways and stimulate cell proliferation [152,153,154,155]. For HCV, NS5A associates with Raf-1 kinase, promoting HCV replication [152]. Consistently, inhibiting Raf kinases with sorafenib blocked the infection, while a further downstream inhibition of MEK1/2 and Erk1/2 showed only marginal effects. This suggests a direct virus–host dependency independent of pathway-associated transcriptional changes. The same holds true for HBV, for which the inhibition of EGFR-associated MAPK or PI3K signaling during infection seems only to have marginal effects [151]. Although no studies have so far demonstrated a role of EGFR during HDV infection, it can be assumed that EGFR may also be required for HDV internalization, since this virus uses the HBV envelope to enter the cell and shares the same entry pathway through HSPG and NTCP.

PI3K/Akt signaling regulates glucose metabolism, cell growth, and survival [156] and it is tightly regulated by phosphatase and tensin homolog deleted on chromosome 10 (PTEN) [157,158]. Independently of its role in insulin signaling, HCV NS5A downregulates PTEN expression through a cooperation of ROS-dependent and -independent pathways that subsequently drives a PTEN–PI3K/Akt feedback loop supporting cell survival [159]. For HBV, the role of PI3K/Akt signaling is more diverse. HBx activates Akt in hepatocytes thereby self-limiting HBV replication [160]. This is consistent with a decreased HBV replication upon PI3K/Akt pathway inhibition using a small-molecule inhibitor in cell culture [161]. However, despite the self-limiting effect on HBV replication, HBx inhibits hepatocyte apoptosis via Akt stimulation and potentially facilitates the persistent, noncytopathic HBV replication [160]. As observed for HCV, HBV impairs PTEN expression, promoting β-catenin/c-Myc signaling and PD-L1 expression [162]. The authors found that PTEN rescue in hepatocytes inhibited β-catenin/PD-L1 signaling and promoted HBV clearance.

Wnt/ß-catenin signaling is essentially involved in the regulation of cell fate during embryogenesis and hepatobiliary development, as well as in liver homeostasis, epithelial–mesenchymal transition (EMT), and tissue regeneration during adulthood. If dysregulated, it promotes liver disease and cancer [163]. The Wnt/β-catenin signaling pathway is activated by hepatic viruses via direct engagement of viral proteins. β-catenin signaling is stimulated by HCV infection via NS3 and NS5A [164] or the phospho-inactivation of GSK-3β by NS5A and core proteins [165,166]. Strikingly, despite a highly genetic heterogeneity, a relative higher frequency of mutations in the β-catenin gene *CTNNB1* can be observed in HCC associated with HCV than with HBV [164]. Wnt/ß-catenin signaling is involved in EMT, which is a hallmark of wound healing and liver fibrosis [167]. During chronic infection, dysregulated wound healing processes cause an excessive deposition of extracellular matrix in the liver, leading to liver fibrosis and cirrhosis, which involve not only hepatocytes but also non-parenchymal cells like HSCs and liver macrophages [167]. The Wnt/ß-catenin cascade has a central role in regulating profibrotic pathways in hepatocytes, which involve oxidative stress signaling and transforming growth factor beta (TGF-β)/SMAD signaling [167]. HCV induces TGF-β signaling indirectly via UPR [168]. Interestingly, TGF-β signaling seems to limit HCV infection in hepatocytes [169]. The virus counteracts this activation by an NS5A-mediated inhibition of the phosphorylation and transcriptional activity of SMAD2 and SMAD3/4 heterodimers [170]. HBV infection is also restricted by TGF-β [171], while HDV seems to stimulate TGF-β in luciferase reporter gene assays [172]. This is consistent with a reported activation of TGF-β expression by HDV via an L-HDAg-mediated activation of the Twist promoter through binding to SMAD3 on Smad-binding elements (SBEs) [173]. This is an interesting finding that may help to understand the aggravation of HBV liver disease and the rapid fibrosis progression in HDV/HBV-infected patients [174].

## 7. Discussion

Chronic liver disease and associated complications including cancer constitute an important burden for public health, with a long-lasting impact on affected individuals even after viral infection cure. The comparison of virus-induced signaling during chronic infection with HBV, HCV, and HDV outlined common pathogenic mechanisms that predominantly result in the failure of the antiviral response to clear infection and in a diversion of the final antiviral safeguard apoptosis towards cell survival. It is evident that the involved signaling pathways that are thereto manipulated largely overlap for different viruses (Table 1), although the detailed strategies differ (Figure 2). Also hepatitis E virus (HEV) seem to dysregulate similar pro-oncogenic signaling pathways linked to oxidative stress, inflammation, apoptosis, and cell proliferation, as reviewed elsewhere [175]. However, chronic infections are relatively rare, and only a fraction of patients progress to fibrosis and HCC [176]. The majority of studies reviewed here were based on cell culture models and performed a limited analysis of canonical pathways. Given these limitations, the currently available literature for some viruses is biased by functional studies of individual proteins (e.g., HBx for HBV) and does not consider protein dynamics or synergic effects of the virus interactome. However, the fast-moving technological development in the recent years and the diffusion of -omics studies in the scientific routine are allowing a more profound study of virus-induced signaling. This should be combined with the use of better infection models representing the three-dimensional architecture of the liver, the heterogeneity of its cell populations, and the contribution of immune cells. Signaling pathways are established targets in cancer therapy [177] and have previously drawn attention as targets for cancer prevention attenuating liver disease progression [178,179]. Host signaling-targeting approaches to battle chronic infection have been discussed [133,180,181] as they hold the potential to lower the genetic barrier of resistance to direct-acting antivirals. However, currently, only interferons are in clinical use targeting chronic viral hepatitis. Thus, a better understanding of virus-induced signaling could promote the development of common therapeutic strategies to help not only patients with chronic infection but also patients suffering from non-viral disease etiologies that display a similar course of liver disease and fibrosis-associated carcinogenesis.

## Figures and Tables

**Figure 1 ijms-23-02787-f001:**
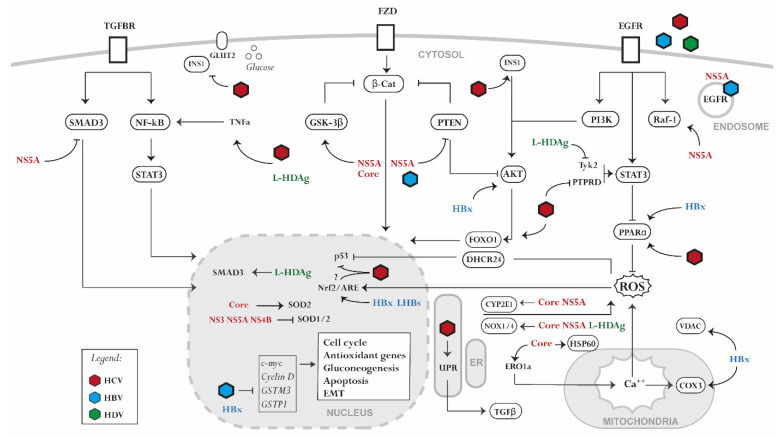
Signaling pathways perturbed by hepatotropic viral proteins. HCV, HBV, and HDV alter liver homeostasis by disrupting several signaling processes associated with (1) the generation of oxidative stress and the dysregulation of the antioxidant system, (2) the alteration of a pro-inflammatory signaling, (3) the hijacking of glucose and lipid metabolism, (4) the dysregulation of host genome expression. NS3, NS4B, NS5A, (non-structural protein 3/4B/5A); HBx, (hepatitis b X antigen); LHBs, (large HBV surface antigen); L-HDAg, (large HDV antigen).

**Figure 2 ijms-23-02787-f002:**
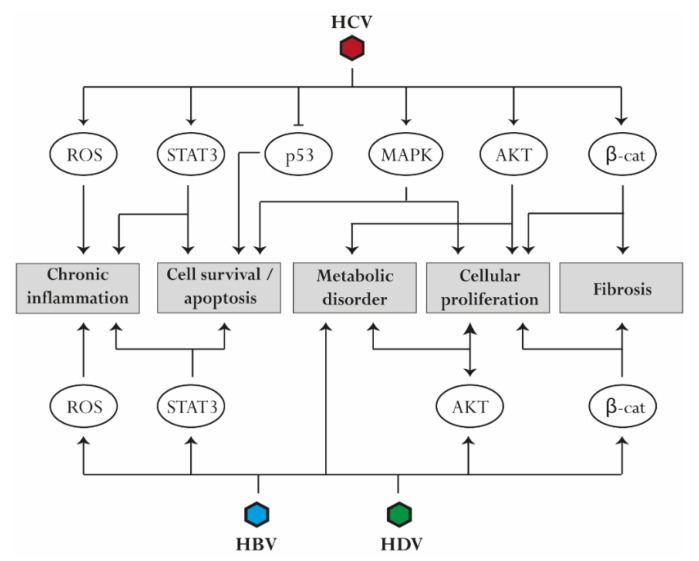
Common pathways associated with virus-induced liver disease progression. Several perturbations are mediated by HBV, HCV, and HDV infection. Reactive oxygen species (ROS) production and activation of STAT3 contribute to the establishment of chronic liver inflammation. Upregulation of mitogen-activated protein kinase (MAPK) and STAT3 signaling as well as downregulation of p53 reduce apoptosis and promote cell survival. Similarly, activation of AKT, MAPK, and β-catenin induces cell proliferation. AKT upregulation contributes to the development of metabolic disorders, while β-catenin is involved in the progression of liver fibrosis.

**Table 1 ijms-23-02787-t001:** Virus-perturbed signaling pathways during chronic viral hepatitis.

Perturbed Signaling Pathway	Virus	References
IL-6/JAK/STAT3	HBV, HCV, HDV	[45,83,89,90,91,92,96]
EGFR	HBV, HCV	[93,94,146,147,148,149,150,151]
TNF-α/NF-κB	HCV, HDV	[45,65,66,82]
Nrf2/ARE	HBV, HCV	[47,48,49,50,54,55,59,60,61]
PI3K/Akt	HBV, HCV	[48,160]
Ras/Raf	HCV	[152]
TGF-β/SMAD	HBV, HCV, HDV	[66,67,69,168,172,173]
Wnt/β-catenin	HBV, HCV	[71,162,164,165,166]

## Data Availability

Not applicable.

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
