# Peer review of "Signaling Induced by Chronic Viral Hepatitis: Dependence and Consequences"

_ijms, 2022, doi:10.3390/ijms23052787_

Round 1

Reviewer 1 Report

This comprehensive review is well written. One major pathological difference of HCV versus HBV infection in liver is associated steatosis/steatohepatitis which determines the disease progression.  Authors mentioned molecular targets/mechanisms but it will be useful to organize in table or diagram.

Author Response

We thanks the reviewer for her/his appreciation of the manuscript. As suggested we now provide a new table 1 summarizing the main signaling pathways altered by the reviewed viruses.

Reviewer 2 Report

Although quite mechanistic, this review is nicely structured, well written and sound in data and style.

in general, this manuscript is of good scientific value and nicely points out common mechanisms of singnaling in chronic viral hepatitis. As I mentioned before, it is quite mechanistic. Therefore it will definitely profit from integrating the shown data into a broader (more clinical based; maybe by some specific examples) context. In addition, it could be improved if the authors will point out potentially existing likewise similarities/associations or at least separating factors regarding their data to rare but existing courses of chronic hepatitis E (HEV) infections.

Minor points:

  • line 71: “ … due to the persistent (=> persistence) of cccDNA pools …”
  • lin 367ff: please rephrase this sentence

Author Response

We thank the reviewer for her/his appreciation of the manuscript and the helpful comments. As suggested, we added a paragraph at lines 390-393 emphasizing similarities to HEV-associated signaling pattern. Moreover, in the revised manuscript we better emphasized the broader clinical perspective in the discussion highlighting the relevance of signaling pathways as targets for antiviral and chemoprevention of liver cancer (lines 415-420).

Minor points:

  • line 71: “ … due to the persistent (=> persistence) of cccDNA pools …”

This has been corrected.

  • line 367ff: please rephrase this sentence

As suggested, we rephrased this sentence to “The Wnt/ß-catenin has a central role in regulating profibrotic pathways in hepatocytes, which involves oxidative stress signaling and transforming growth factor beta (TGF β)/SMAD signaling [167].”